# Electrophysiology of hiPSC-Cardiomyocytes Co-Cultured with HEK Cells Expressing the Inward Rectifier Channel

**DOI:** 10.3390/ijms22126621

**Published:** 2021-06-21

**Authors:** Ana Da Silva Costa, Peter Mortensen, Maria P. Hortigon-Vinagre, Marcel A. G. van der Heyden, Francis L. Burton, Hao Gao, Radostin D. Simitev, Godfrey L. Smith

**Affiliations:** 1Institute of Cardiovascular and Medical Sciences, College of Medical, Veterinary and Life Sciences, University of Glasgow, 126 University Place, Glasgow G12 8QQ, UK; Ana.Costa@glasgow.ac.uk (A.D.S.C.); francis.burton@glasgow.ac.uk (F.L.B.); 2School of Mathematics and Statistics, University of Glasgow, Glasgow G12 8QQ, UK; peter.mortensen@glasgow.ac.uk (P.M.); Hao.Gao@glasgow.ac.uk (H.G.); Radostin.Simitev@glasgow.ac.uk (R.D.S.); 3Department of Biochemistry and Molecular Biology and Genetics, College of Sciences, University of Extremadura, 06006 Badajoz, Spain; mahortigonv@unex.es; 4Department of Medical Physiology, Division of Heart & Lungs, University Medical Center Utrecht, Yalelaan 50, 3584 CM Utrecht, The Netherlands; m.a.g.vanderheyden@umcutrecht.nl

**Keywords:** hiCMs, HEK, I_K1_, electrophysiology, co-culture, maturation

## Abstract

The immature electrophysiology of human-induced pluripotent stem cell-derived cardiomyocytes (hiCMs) complicates their use for therapeutic and pharmacological purposes. An insufficient inward rectifying current (I_K1_) and the presence of a funny current (if) cause spontaneous electrical activity. This study tests the hypothesis that the co-culturing of hiCMs with a human embryonic kidney (HEK) cell-line expressing the Kir2.1 channel (HEK-I_K1_) can generate an electrical syncytium with an adult-like cardiac electrophysiology. The mechanical activity of co-cultures using different HEK-I_K1_:hiCM ratios was compared with co-cultures using wildtype (HEK–WT:hiCM) or hiCM alone on days 3–8 after plating. Only ratios of 1:3 and 1:1 showed a significant reduction in spontaneous rate at days 4 and 6, suggesting that I_K1_ was influencing the electrophysiology. Detailed analysis at day 4 revealed an increased incidence of quiescent wells or sub-areas. Electrical activity showed a decreased action potential duration (APD) at 20% and 50%, but not at 90%, alongside a reduced amplitude of the aggregate AP signal. A computational model of the 1:1 co-culture replicates the electrophysiological effects of HEK–WT. The addition of the I_K1_ conductance reduced the spontaneous rate and APD20, 50 and 90, and minor variation in the intercellular conductance caused quiescence. In conclusion, a 1:1 co-culture HEK-I_K1_:hiCM caused changes in electrophysiology and spontaneous activity consistent with the integration of I_K1_ into the electrical syncytium. However, the additional electrical effects of the HEK cell at 1:1 increased the possibility of electrical quiescence before sufficient I_K1_ was integrated into the syncytium.

## 1. Introduction

Human-induced pluripotent stem cell-derived cardiomyocytes (hiCMs) are a potential alternative to adult cardiomyocytes from animal models for pharmacotoxicity and regenerative medicine [1,2,3,4,5,6]. This cell source could circumvent inter-species differences and provide human-translatable information [4,7]. hiCMs have been widely studied in recent years, and their electrical phenotype has been described as immature [3,7]. This has been attributed in part to the expression of the pacemaker current (if) in hiCMs that are nominally ventricular in their electrical phenotype [1,7,8]. Furthermore, hiCMs lack inward rectifying current, I_K1_ [1,3,9,10,11], a conductance that establishes the resting membrane potential in ventricular cardiomyocytes [10,12,13,14,15]. Strategies to overcome the lack of I_K1_ in hiCMs include the supplementation of ionic I_K1_ current with an electrical current of similar characteristics [7,16,17], but this cannot be applied to a monolayer or 3D culture. Alternatively, a transfection with the I_K1_ protein can result in stable membrane potentials, but regulating the level of the expressed protein, and therefore the electrophysiological effect, is difficult [18].

There is precedent for the use of a co-culture of two cell lines in order to influence the overall electrophysiological behavior of the syncytium [19,20]. This technique has the potential to introduce ionic conductance into the syncytium in a controlled manner, as the relative number of cells can be easily controlled. In this study, the co-culture of hiCMs with HEK cells expressing I_K1_ is investigated as a method to introduce I_K1_ function into a syncytium of hiCMs, in order to reproducibly alter electrophysiology and create a more adult-like ventricular action potential. The HEK-293 cell line is a suitable choice for electrophysiological studies for several reasons; firstly, it is a cell line that does not express significant amounts of selective ion channels and has a near-zero membrane potential [21]. Secondly, the cell line expresses the connexin 43 hemichannel, which facilitates electrical coupling to adjacent cells [22].

The aim of this study is to investigate the effect of supplementing I_K1_ on the electrophysiology of hiCMs via co-culture.

## 2. Results

### 2.1. Effects of Co-Culture on Spontaneous Rate and Contraction Duration

Assuming the frequency of spontaneous electrical activity can be used as an assay of the influence of I_K1_ activity on coupled HEK cells, the spontaneous contractile activity of the co-culture of hiCMs with HEKs was investigated at different ratios of HEK:hiCM (Figure 1A). When the ratio of either HEK-I_K1_:hiCM or HEK–WT:hiCM was increased from 1:30 (Figure 1(Bi)) to 1:10 (Figure 1(Bii)), there was no change in spontaneous frequency throughout the 8 days of incubation. At a ratio of 1:3 on day 5, there was a decrease in average frequency in the presence of HEK-I_K1_ (Figure 1(Biii)), decreasing from 1.29 ± 0.06 Hz on day 4 to 1.13 ± 0.05 Hz on day 5, and a further decrease to 1.07 ± 0.05 Hz on day 6, 0.96 ± 0.05 Hz on day 7, and 1.00 ± 0.06 Hz on day 8. The largest effect due to the presence of HEK-I_K1_ cells was seen at the maximum ratio (1:1), where the frequency was decreased significantly from day 3 to day 8 (Figure 1(Biv)). On day 3, the frequency was 0.84 ± 0.07 Hz, compared to the hiCM at 1.39 ± 0.08 Hz. This decreased further on day 4 to 0.67 ± 0.04 Hz—the contractility videos illustrating the differences in spontaneous rate are shown in the online Appendix A. The co-incubation of hiCMs with the HEK–WT cell line also caused a decrease in spontaneous rate, but to a considerably lower extent. To illustrate these effects, the change in the mean frequency was expressed relative to the frequency of hiCMs at day 4, day 6 and day 8 (Figure 1C). The data show that mean frequency was decreased in a ratio-dependent manner in the HEK-I_K1_:hiCM co-culture, with a maximum effect seen at a ratio of 1:1 on day 4 (Figure 1(Ci)). Similar but smaller effects were seen on day 6 (Figure 1(Cii)) and day 8 (Figure 1(Ciii)). The analysis of the time course of contraction (CD50) (Figure 1D) revealed no consistent functionally significant change in any of the groups across the incubation period, nor was there any change in the average amplitude signal (data not shown). Based on these data, the incubation time was subsequently standardized at day 4 to allow a more detailed investigation of the contraction and electrophysiology.

### 2.2. Effect of Co-Culture at Day 4 on Gross Contraction Characteristics

Figure 2 shows the data for day four co-cultures that were transferred to serum-free media. Example contractility records are shown in Appendix A. As shown in Figure 2A, a co-culture with HEK–WT produced small effects on the spontaneous rate (less than 20%). The frequency decreased to 0.95 ± 0.01 Hz in HEK–WT:hiCM vs. 1.12 ± 0.02 Hz in hiCM (Figure 2A).

The co-culture with HEK-I_K1_ produced a further decrease in frequency (approx. 50%), and the variability in the spontaneous rate was increased. The average frequency (0.59 ± 0.03 Hz) was 58% of hiCM controls (Figure 2A and Appendix A). The amplitude of the contraction (Figure 2B) was on average larger, but more variable in HEK-I_K1_:hiCM (spontaneous: 1572 ± 95 a.u.; 1 Hz: 1440 ± 98 a.u.; 2 Hz: 1449 ± 83 a.u.) than in HEK–WT:hiCM (spontaneous: 1155 ± 69 a.u.; 1 Hz: 1216 ± 86.44 a.u.; 2 Hz: 1092 ± 70 a.u.) and hiCM (spontaneous: 815 ± 50 a.u.; 1 Hz: 796 ± 51 a.u.; 2 Hz: 764 ± 46 a.u.) (variability displayed in Appendix A). The time course of contraction was relatively unaffected by the co-culture, as shown in Figure 2C–E. Time to contract was reduced in HEK–WT:hiCM at 1 Hz (60 ± 1 ms) compared to hiCM (74 ± 5 ms) (Figure 2(Cii)). The only variable that showed significant change was time to relax in HEK-I_K1_:hiCM, which increased to 504 ± 33 ms (vs. HEK–WT:hiCM: 267 ± 11 ms vs. hiCM: 230 ± 9 ms) during spontaneous activity. A difference between the groups was not seen during stimulation (1 Hz or 2 Hz) (Figure 2E). The individual data points are presented in Appendix A.

### 2.3. Effect of Co-Culture at Day 4 on Spatial Characteristics of Contraction

The spatial information on cell movement was analyzed by creating a 10 × 10 grid from the 200 × 200 µm area of contracting cells. The range of movement amplitudes was assessed individually for the 100 sub-areas, as was the percentage of active sites and average amplitude (Figure 3A). One difference noted was the functionally small changes in the percentage of active sites across all three groups, but only the HEK-I_K1_:hiCM group displayed a significant number of sites that were mechanically inactive (zero active sites); see Appendix A. These inactive sites were not present during stimulation (1 Hz and 2 Hz). The range of movement amplitudes within each 10 × 10 grid is shown in Figure 3D, indicating that the highest variability across areas was seen in spontaneously active HEK-I_K1_:hiCM co-cultures (Appendix A).

### 2.4. Effects of Co-Culture at Day 4 on Total Cellular Fluorescence after FluoVolt Staining

The voltage sensitive dye binds to the membrane outerleaflet and has a large baseline fluorescence and smaller dynamic effects due to voltage changes (<10%). The total fluorescence reflects the membrane area stained by the dye, which in turn reflects the number of cells in a specified area. The total fluorescence due to FluoVolt was used to indicate the total area of membrane stained by the dye. The fluorescence amplitude (Appendix A) was larger in the co-culture, where the value obtained in HEK-I_K1_+hiCM was 4504 ± 116 photon/s (2 × 25,000 cells/well) and 4614 ± 104 photon/s in HEK–WT+hiCM (2 × 25,000 cells/well), compared to hiCM where the average fluorescence was 3680 ± 83 photon/s (25,000 cells/well see Appendix A). The amplitude of the action potential signal was smaller in 1:1 HEK-I_K1_+hiCM (36.5 ± 5.1 photon/s) compared to 1:1 HEK–WT (77.2 ± 3.9 photon/s) and hiCM (96.0 ± 5.0 photon/s) (Appendix A). The causes of the differences in the AP amplitude signal are not known, and may be partially due to the average signal derived from the electrically inactive and active cells types, and partially due to the genuine differences in the action potential amplitude of hiCMs in the co-culture. Interference/contamination of the FluoVolt fluorescence with GFP fluorescence in the HEK-I_K1_ group is unlikely, since both HEK groups gave similar total fluorescence values. This may be due to the weak GFP signal and sub-optimal excitation and emission wavelengths.

### 2.5. Effect of Co-Culture at Day 4 on Action Potential Waveform Characteristics

Immediately after contractility, fluorescent signals were recorded from a 200 × 200 µm area of co-cultured cells previously treated with voltage-sensitive fluorescent dye (FluoVolt™). Example fluorescence traces (Appendix A) show the effects of co-culture with HEK–WT (panel i) and HEK-I_K1_ (panel ii) on the time course and magnitude of fluorescent change relative to diastolic fluorescence (F0).

As with Figure 4, the averages from four separate platings for each plate are plotted (±SD). In general, the co-culture with HEK–WT has minimal effects on spontaneous rate, but reduces the APD at the early-, mid- and late-repolarization phases. This is evident in both the spontaneous and stimulated preparations. The average magnitude was reduced by 10–20% in APD. Co-culturing with HEK-I_K1_ cells displayed a limited range of effects. In two experiments, the co-culturing caused a significant reduction in spontaneous frequency (experiment 1 and 3), while in two other experiments, the spontaneous frequency was not altered. Similarly, the APD was reduced at APD20 and 50 in experiments 1 and 3, and was not altered significantly in experiments 2 and 4. At APD90, the durations were prolonged in all experiments when compared to HEK–WT.

The early repolarization phase (APD20) was decreased by 25% compared to control values (100% hiCM culture) at the spontaneous rate (HEK-I_K1_:hiCM: 74.2 ± 4.2 ms and HEK–WT:hiCM: 81.0 ± 3.2 ms vs. hiCM: 126.6 ± 4.5 ms) and stimulation with 1 Hz (HEK-I_K1_:hiCM: 64.9 ± 4.7 ms and HEK–WT:hiCM: 77.7 ± 4.1 ms vs. hiCM: 107.8 ± 3.8 ms), 2 Hz (HEK-I_K1_:hiCM: 66.2 ± 4.2 ms and HEK–WT:hiCM: 71.4 ± 3.6 ms vs. hiCM: 103.7 ± 3.4 ms) and 3 Hz (HEK-I_K1_:hiCM: 63.0 ± 5.2 ms and HEK–WT:hiCM: 65.0 ± 3.1 ms vs. hiCM: 89.6 ± 2.2 ms) (Figure 4C). The APD20 was significantly shorter in HEK-I_K1_:hiCM at 1 Hz and in HEK–WT:hiCM (ms). This effect on AP shape was still present at APD50 (Figure 4D). the HEK–WT:hiCM APD50 shortened at all rates (HEK–WT:hiCM spontaneous 138.1 ± 3.9 ms, 1 Hz: 134.2 ± 4.0 ms, 2 Hz: 122.9 ± 4.6 ms and 3 Hz: 109.8 ± 4.1 ms), compared to hiCM (spontaneous: 206.1 ± 6.3 ms, 1 Hz: 185.0 ± 6.1 ms, 2 Hz: 175.1 ± 4.3 ms and 3 Hz: 143.7 ± 3.0 ms) (Figure 4D). In HEK-I_K1_:hiCM, the APD50 was significantly shorter (spontaneous: 146.2 ± 7.4, 1 Hz: 146.6 ± 8.2, 2 Hz: 139.4 ± 6.4, 3 Hz: 123.8 ± 5.9 ms). The APD50 was significantly longer at 2 Hz and 3 Hz in HEK-I_K1_:hiCM compared to HEK–WT:hiCM. Figure 4E shows the effects on APD90, which was shorter in HEK–WT:hiCM compared to hiCM (spontaneous: 211.8 ± 8.9 ms vs. 269.7 ± 6.2 ms, 1 Hz: 212.3 ± 9.5 vs. 256.1 ± 5.4 ms, 2 Hz: 192.4 ± 7.5 vs. 232.2 ± 4.2 ms, 3 Hz: 170.3 ± 6.3 vs. 200.0 ± 2.1 ms). The APD90 was shorter in HEK-I_K1_:hiCM compared to hiCM, but longer than that seen in HEK–WT:hiCM at spontaneous rates (248.7 ± 11.2 ms). At different pacing frequencies, APD90 was prolonged (1 Hz: 267.9 ± 11.4 ms, 2 Hz: 241.3 ± 10.1 ms, 3 Hz: 228.1 ± 7.0 ms) compared to other cultures.

### 2.6. Computational Modelling of HEK:hiCM Coupling

A computational model was developed using published equivalent circuits for HEK [21] and hiCMs [17,23,24], linked by a resistance representing the gap junction (Figure 5A). The presence of connexin43 in both cell types has been confirmed previously [22,25], and immunohistochemistry has confirmed the connexin43 expression in both the HEK–WT and HEK-I_K1_ cell lines used in this study (Appendix A). Example traces are shown in Figure 5B; the isolated hiCM is shown in comparison with the hiCM linked to an HEK cell and the corresponding hiCM as a result of the electrical coupling to an HEK cell with a G_gap_ of 0.7 nS. The dependence of the action potential characteristics on the coupling conductance between the HEK cell and hiCM is shown for two forms of the hiCM model: (i) Paci [17] and (ii) the Paci model with half the specified sodium conductance (0.5 × INa). Increasing the gap junction conductance had minimal effects on frequency across a wide range of values, but caused a small decrease in spontaneous frequency (similar to experimental values) in the 0.5 × I_Na_ group. Both groups failed to generate spontaneous activity above the threshold of 1.7–2 nS G_gap_. The main effect of coupling to HEK is a progressive shortening of APD 20, 50, and 90. Based on the experimental data, which showed a decrease in APD at 1:1 of approximately 20%, an estimate of the value of G_gap_ that would generate an effect of similar magnitude in the computational model was estimated (0.7 nS), as shown by the dotted line in Figure 5C. In summary, using the changes in APD observed experimentally, the in silico model allows an estimate of the gap junction conductance (G_gap_) that would explain the changes in APD observed. This estimate was used for the next stage of the computational modeling, namely, the incorporation of I_K1_ conductance in the coupled HEK cell model.

In Figure 6, the effect of adding I_K1_ into the cell syncytium was modeled by using the published I–V curve for this cell line [7,26]. Example traces are shown in Figure 6B. In comparison with Figure 5B, the HEK cell component has a more polarized membrane potential, closer to the potassium reversal potential (E_K_). As with Figure 5, the influence on the electrophysiology of the hiCM was examined for two versions of the Paci model: (i) normal and (ii) 0.5 × I_Na_. As the value of I_K1_ is scaled from zero to the measured value (3 S/F) and beyond, the minimum potential dramatically decreased to −75 mV and the spontaneous frequency was decreased, albeit by relatively small amounts, in both forms of the model. APD20, 50 and 90 all decreased. These results are consistent with the experimental data in all but one respect: the APD90 value did not increase, and in most instances increased upon addition of I_K1_. The reason for this disparity is not clear.

In Figure 6D, the sensitivity of the two systems (HEK–WT:hiCM, HEK-I_K1_:hiCM) to the value of G_gap_ was explored by using the published G_IK1_ value of 3S/F [7] and varying G_gap_ levels from 0 to 2 nS. As seen in Figure 5, increasing G_gap_ caused changes in spontaneous frequency combined with a progressive decrease in APD. This picture was similar for the 0.5 × I_Na_ Paci model, but at G_gap_ levels greater than approximately 1 nS, the spontaneous activity was abruptly curtailed.

## 3. Discussion

This study was designed to investigate a method for reproducibly introducing a specific conductance (I_K1_) into a hiCM syncytium so as to generate an overall electrical behavior that would facilitate the use of hiCM tissues in cardio-toxicological screening and regenerative medicine. Commercial sources of hiCMs consist of a cell population with a broad range of electrophysiological phenotypes [27], which cover nodal/atrial/ventricular phenotypes of adult hearts [28]. These different electrophysiological phenotypes are not obvious in a monolayer culture because the strong electrical coupling between cells via connexin junctions ensures an action potential waveform that is an average of all contributing cells. A feature of all hiCMs is spontaneous electrical activity due to a slowly depolarizing diastolic potential, which is thought to be a consequence of the significant expression of the funny current (I_f_) [1,7,8] and the lower expression of the I_K1_ current in these cells [1,3,9,10,11]. The effect of the resultant diastolic pacemaker potential in phase 4 of the action potential is the incomplete recovery of I_Na_ from inactivation, leaving less current available to support phase zero. Insufficient I_K1_ will also reduce the outward current contribution to the repolarizing phase, prolonging the APD. Prior strategies to improve hiCMs by enhancing I_K1_ activity have limitations, including transfection-mediated increased IK1 expression, where transfection efficiency is difficult to control [18].

Co-cultures of hiCMs with HEK-I_K1_ cells were generated to attempt to introduce a reproducible I_K1_ function to a syncytium of hiCMs. Mechanical function was studied by monitoring cell motion, and electrical function was assessed using a fluorescent voltage-sensitive dye. Ratios of HEK:hiCM of 1:30 and 1:10 did not cause significant changes in spontaneous rate or contraction time of the hiCMs at any point in the incubation period (8 days). A ratio of 1:3 HEK-I_K1_:hiCM significantly reduced the spontaneous rate from day 5 to day 8. The ratio 1:1 HEK-I_K1_:hiCM significantly reduced the rate from day 3 onwards with a maximum effect on day 4. Note, there was no change in the duration of contraction throughout the culture period as spontaneous frequency changed. This reinforces the point that the change in frequency specific to the HEK-I_K1_ co-culture is due to a change in the intrinsic pacemaker, and not due to changes in the time course of contraction. These results suggest that the co-culture of hiCM with HEK-I_K1_ had a maximal functional effect in terms of spontaneous frequency on days 4–6 of co-culture, which diminished after day 6. The use of an altered spontaneous rate as an index of additional I_K1_ function is supported by the modeling of the electrophysiology of hiCM (Appendix A). As the I_K1_ function is increased from a baseline of 25 nS, there is a gradual reduction in spontaneous frequency and APD, reaching a critical value at ~60 nS per cell, beyond which the spontaneous rate steeply decreases until complete quiescence at approximately 65 nS I_K1_ per cell. The slowing of spontaneous rate effect due to the use of HEK-I_K1_ cells is consistent with studies introducing an inward rectifying current [7] and transfection of I_K1_ [18]. In Bett et al. [7], the introduction of an I_K1_-like current electronically into a single hiPSC-hiCM caused a significant shortening of APD, and a slowing of the spontaneous rate [29]. Alternative explanations for this effect, e.g., the decreased expression of the funny current (I_f_) in hiCMs selectively co-cultured with the HEK-I_K1_ cell line and not with HEK–WT, seem unlikely, but have not been tested directly.

### 3.1. Effect of Co-Culture of hiCM with HEK–WT

Based on the initial co-culture experiments, the 1:1 culture at day 4 was chosen as the optimum point to study the contractile and electrical consequences of the hiCMs–HEK–I_K1_ system. Under these conditions, the control co-culture of 1:1 HEK–WT:hiCM was not without effects on contractile and electrical parameters, including a small but significant decrease in spontaneous rate. These contractile effects were retained in serum-free solution and the associated spatial contractile analysis (Figure 3) indicates that the amplitude and range of contraction amplitudes over the spatial grid were increased (by 10–20%). The reasons for this are unclear, but it could be a feature of the higher cell density in the co-culture changing the mechanical properties of the monolayer, or simply an increase in the number of contrasting features within the image.

In terms of electrophysiology, the co-culturing of hiCM with HEK–WT caused a similarly small (~10%) decrease in spontaneous rate, but a larger decrease (~20%) in APD. It is unlikely that the small decrease in spontaneous rate is a consequence of background expression of I_K1_ in HEK–WT cells, as there is no electrophysiological evidence of this conductance in the WT cell [21]. Computational electrophysiological modeling of 1:1 cultures of hiCM and HEK cells showed progressive decreases in APD when gap junction conductance is increased [30]. Although the connexin 43 hemichannel is expressed in HEK–WT cells [22] and their expression was confirmed in this study using immunohistochemistry (Appendix A), the effective gap junction resistance is unknown. Using the experimental data on APD, a comparable change in APD was achieved when the coupling conductance (G_gap_) had a value of approximately 0.7 nS. The modeling showed that during the plateau of the AP, the HEK generated a depolarizing current, which caused a more rapid repolarization of the hiCM. In these respects, the experimental data and computational model are in close correspondence and establish an estimate of G_gap_. Two forms of the cardiomyocyte model were used in the study: the specified Paci model [17,23], and another using this model but with a reduction in I_Na_ conductance by 50%. The modified model was used to investigate the effects of sodium current (I_Na_) on the behavior of the coupled model. In both scenarios, increasing the coupling to HEK–WT reduced APD, but the simulation was unable to generate spontaneous activity at a G_gap_ of approximately 2 nS. As shown in Figure 5, the cause of the loss of AP signal is due to the depolarization of the paired cell system and the loss of excitability. Note that in none of the experimental co-cultures involving HEK–WT was quiescence observed (Appendix A); this supports the estimate of G_gap_ being <2 nS, and based on the APD changes observed, the value of 0.7 nS was estimated.

The amplitudes of total fluorescence (Fo) from Fluovolt provide information on the total amount of membrane accessible to the dye. As shown in Appendix A, the addition of 25,000 HEK–WT to 25,000 hiCM increased the membrane fluorescence signal from 4499 ± 169 photons/s to 5792 ± 230 photons/s. Assuming the FluoVolt dye is partitioned equally between both cell types, double the cell number would be expected to increase the total signal by an additional ~4500 photons/s; the increase by ~1300 photons/s suggests that the overall surface area of the HEK cells is only 30% of the hiCM.

### 3.2. Effect of Co-Culture with HEK-I_K1_ on Electrophysiology of hiCM

At the same day 4 time point, the use of HEK-I_K1_ cell line significantly decreased the spontaneous rate (to ~50%) but had little effect on gross contractile parameters (Figure 4). Spatial analysis of the contractile activity (Figure 3) showed that in the 1:1 HEK-I_K1_:hiCM co-culture, there was a significant numbers of wells with large areas of inactive cells, and in some wells, no activity was seen, i.e., mechanical quiescence. Furthermore, there was an increased range of contraction amplitudes across the surface of the culture (Figure 3). These data indicate that although gross contractility was relatively unaffected in the HEK-I_K1_–hiCM group, there was considerably greater heterogeneity within the culture with areas that are not spontaneously active (but responded to stimulation) and areas with widely disparate contractility amplitudes, suggesting a greater electrical heterogeneity in the HEK-I_K1_–hiCM co culture. These two effects may be related, as mechanically quiescent areas of cells coupled to active cell areas may generate larger movements within the culture. Measurements of the mean changes in spontaneous rate across multiple plates varied between the four experiments. Similarly, when studying the extent of the effects on AP time course, this effect also varied. In general terms, APD20 and 50 decreased, but in none of the measurements did APD90 decrease significantly, and in some cases APD90 was increased compared to the HEK–WT co-culture. The reason for this disparity is unknown; one potential over-simplification of the model is the linear background current and reversal potential attributed to the HEK cell. Although this is based on published data, the real electrophysiology is more complex, with both individual outward and inward currents [31], so future studies should include a more detailed electrophysiological model.

Adding I_K1_ to the HEK within the computational model caused parallel decreases in APD at 20, 50 and 90% repolarization, with values in some platings of the HEK–WT reaching approximately 95%. With a G_gap_ of 0.7 nS, no electrical instability was observed. However, when the G_gap_ value was increased marginally from 0.7 to 1 nS, the myocyte model with 0.5 × I_Na_ became electrically quiescent. This suggests that spontaneous activity in the 1:1 HEK:hiCM system may be very sensitive to the G_gap_ value and/or precise ratio of hiCMs to HEK cells. This behavior seen only in the 0.5 × I_Na_ system mimics the increased tendency towards electrical quiescence seen in the experimental systems. Quiescence in this version of the model appeared at a lower gap junction conductance (1 nS, Figure 6C) when compared to the equivalent HEK–WT model (2 nS, Figure 5C), and was not related to the depolarization of the cell pair. The system remained polarized at close to −80 mV. This indicates that by reducing the excitability of the Paci model, the cell pair model reproduces the electrical/mechanical quiescence seen in some co-cultures with HEK-I_K1_:hiCM. This supports the concept that substantial fractions of the co-cultures are electrically quiescent in HEK-I_K1_:hiCM co-cultures. This probabilistic behavior is difficult to model computationally, but is evident in the sensitivity of the model to small increases in G_gap_. In experimental terms, it can be explained by small variations in the electrophysiology of the hiCM culture, including the magnitude of ionic conductance that determines excitability. The large areas of electrically unexcitable hiCMs explains the variability in contraction amplitude and the smaller amplitude voltage signal observed experimentally, since a fraction of the co-culture will not produce action potentials. The total baseline fluorescence signal from the HEK-I_K1_:hiCM co-culture was not significantly different from that seen in HEK–WT:hiCM, and supports the idea that a comparable number of cells were involved in both co-culture situations.

### 3.3. Comparing Enhanced I_K1_ vs. Coupling to HEK-I_K1_

Increases in I_K1_ in the Paci hiCM model (Appendix A) caused electrophysiological changes that were similar in general, but differed in the details of the experimental and modeling data for the 1:1 HEK-I_K1_:hiCM system. An increase in I_K1_ by 10 S/F in the hiCM model caused only a small change in spontaneous rate, but a significant shortening of APD30, 50 and 90. The spontaneous frequency was reduced to half by enhancing I_K1_ by 30 S/F. In contrast, coupling 3 S/F I_K1_ via the HEK cell reduced the spontaneous frequency, with significant effects on APD20, 50 and 90. Therefore, in terms of the amount and the details of the effects, adding I_K1_ to the cell and adding it via a co-culture system produce far from identical effects. The major influence on hiCM electrophysiology in the 1:1 syncytium with HEK-I_K1_ is due to the coupling of hiCMs to a network of HEK-I_K1_ cells that are strongly polarized to approx. −75 mV. Under these conditions, the APD values of the hiCMs are shortened due to the current flowing to the polarized HEK-I_K1_ cell network, and this explains the seemingly large effect of the small I_K1_ conductance in the HEK-I_K1_ cells. This suggests that the HEK-I_K1_ cell line does not have suitable electrophysiological characteristics to introduce I_K1_ to the hiCM via this approach. Furthermore, the variability in the magnitude of the effect between different platings suggests that these effects are not reproducible; the reasons for this are unclear, but may reflect differing extents of expression of I_K1_ in different passages of the HEK-I_K1_ cell line. An additional feature is the polycolonal nature of the HEK-I_K1_ cell line, which means there is significant variability in the expression level of I_K1_ per cell. The selection of subpopulations of high-expression cells (via GFP fluorescence) before co-culture is a strategy that has been used to overcome this [28].

In conclusion, this study shows that viable co-cultures and electrical coupling can be achieved between hiCM and HEK cells with minimal changes in the electrophysiology of the overall syncytium. However, the use of this system to introduce specific ionic conductances, in this case I_K1_, was limited because the coupling of a significant amount of I_K1_ results in areas of culture that are electrically and mechanically quiescent, a feature that was overcome by field stimulation. The reason for this non-homogeneous effect is not known, but may represent the electrical heterogeneity in both hiCM and HEK-I_K1_, along with heterogeneous coupling between both cell types. The conclusion of this study is that introducing I_K1_ by this method required a 1:1 ratio of the two cell types. This number of electrically quiescent polarized HEK cells coupled to hiCMs will inevitably introduce regions of reduced excitability and blocked conduction before sufficient I_K1_ conductance can be introduced to the syncytium. This shortcoming can only be addressed by using partner cells with substantially greater (10×) I_K1_ than the current HEK-I_K1_ cell line. The issues over the reproducibility of the approach need to be addressed separately.

## 4. Materials and Methods

### 4.1. Co-Culture with hiCMs and HEK

We coated 96-well glass-bottom plates (MatTek, Maddison, WI, USA) with 10 µg/mL fibronectin from bovine plasma (F1141, Sigma-Aldrich, Germany) prepared in Dulbecco’s phosphate buffered saline (DPBS) with MgCl_2_ and CaCl_2_ (D8662, Sigma-Aldrich, Germany). Plates were incubated for 3 h at 37 °C or overnight at 4 °C. Cor.4U hiPSC-CMs (hiCM) (NCardia, Leiden, The Netherlands) were thawed as per manufacturers’ instructions in Cor.4U medium. Two lines of HEK293 were used: HEK293 transfected with pcDNA3-Kir2.1-GFP (I_K1_), which stably express I_K1_ [26], and wild-type HEK293 (WT). HEK293 were thawed in Cor.4U media. Manual cell counts were performed for hiCM and HEK and cell suspension densities were adjusted. The hiCM cell density was adjusted to 250,000 cells/mL and 25,000 cells were plated per well to ensure monolayer formation. Different densities were tested: 1:30, 1:10; 1:3 and 1:1 HEK:hiCM. Cor.4U were thawed and a cell count was performed; cell density was adjusted to 125,000 cells/mL, and the cells were transferred to separate sterile tubes so that different densities of HEK could later be added and mixed. HEK293, both WT and I_K1_-expressing, were thawed in HEK medium, centrifuged at 1050 rpm for 2 min, and then resuspended in Cor.4U maintenance medium, at which point a cell count was performed and the cell density was adjusted to 125,000 cells/mL. Suspensions of hiCMs and HEK cells were mixed to achieve the different ratios of HEK:hiCM and plated conventionally, maintaining 25,000 hiCMs/well with different accompanying numbers of HEK cells.

### 4.2. Contractility and Voltage Recordings

Contractility was recorded daily after plating in an environmentally controlled stage incubator at 37 °C, with 5% CO_2_ and 75% humidity, using a 40× 0.6NA Olympus objective via HCImage Live (Hamamatsu Corporation, Bridgewater, NJ, USA) controlling a Hamamatsu ORCA-Flash4.0 V3 Digital CMOS camera at 100 fps (2048 H × 2048 V pixel resolution). Short periods (15–20 s) of video were analyzed using a ContractilityTool application (developed by F.L.Burton) which is an implementation of the MUSCLEMOTION algorithm [32]. This application derives several parameters from the resultant motion signal including the average spontaneous frequency, amplitude, the duration at 50% of the contraction transient amplitude (CD_50_), the time to contract (from 10% to 90% of the contraction amplitude) and time to relax (time from 90% to 10% of the contraction amplitude). While this analysis is normally reported for the whole image, this application also allows the video image to be subdivided into a grid of images (e.g., 10 × 10 grid), and these 100 images were individually analyzed for motion to provide a spatial dimension to the analysis, as shown in Figure 3.

On the last day of culture, monolayers were loaded with the voltage-sensitive dye (VSD) FluoVolt at 1:1000 and the membrane protectant, PowerLoad, at 1:100 (F10488, ThermoFisher, UK), diluted in the serum-free medium, BMCC (NCardia, The Netherlands). Cultures were incubated for 25 min at 37 °C, in 5% CO_2_. Voltage was recorded at a 10 kHz sampling rate. The VSD was excited at 470 nm, and emission was recorded using a photomultiplier at wavelengths longer than 535 nm [32,33]. Data were acquired and subsequently analyzed on CellOPTIQ software (Appendix A) (Clyde Biosciences Ltd, Newhouse, Scotland, UK).

The spontaneous electrical and therefore mechanical activity of the hi-CM cultures was over-paced by external stimulation provided by a pair of custom-made graphite pin-electrodes ~4 mm apart lowered into the media, and a voltage pulse (<40 V, 6 ms duration) was provided by a IonOptix LLC (Westwood, MA, USA) C-pace stimulator unit.

### 4.3. In Silico Modelling of Coupled HEK-Cardiomyocyte Electrophysiology

Previous work by MacCannell et al. [34] coupled the electrophysiological models of cardiomyocytes and fibroblasts. Based on this approach, a novel model of HEK cell electrophysiology was coupled with an established model of hiCM electrophysiology [17,22,23]. The transmembrane potentials of the two cell types are coupled using a resistive term (1/*G_gap_*). Thus, the strength of this coupling is dependent on the *G_gap_*, and the number of hiCM per HEK cell (*n*). The value of n is used to define the ratio of hiCM to HEK cells, for example, if *n* = 30, the hiCM:HEK cell ratio is 30:1. To go below a 1:1 ratio, i.e., more hiCMs than HEK cells, the fractional n is used, so for a hiCM to HEK cell ratio of 1:2 then *n* = 1/2. HEK cell electrophysiology is simplified to 2 currents: (i) a background linear conductance with a reversal potential of −30 mV [21,35] with no evidence of inward rectification, and (ii) an I_K1_ conductance, added to the background current in the HEK-I_K1_ simulations, wherein the magnitude was based on experimental measurements [26]. The mathematical description of I_K1_ current was represented as previously published [36] and fitted to the data by de Boer et al. [26]. The primary Equations (1)–(4) used in the model are listed in below:(1)$$C_{m}\frac{dV_{m}}{dt}=-\left(I_{m}+G_{\mathrm{gap}}\left(V_{m}-V_{H}\right)\right),$$
(2)$$C_{H}\frac{dV_{H}}{dt}=-\left(I_{\mathrm{WT}}+I_{K1}+nG_{\mathrm{gap}}\left(V_{H}-V_{m}\right)\right),$$
(3)$$I_{\mathrm{WT}}=0.0726\left(V_{H}+30\right),$$
(4)$$I_{K1}=2.9710\frac{V_{H}+78.3651}{1+2.5156e^{\frac{V_{H}+65.8558}{12.9710}}}.$$

Here, *V_m_* and *V_H_* are the myocyte and HEK cell transmembrane potentials, respectively (with units mV), *C_m_* and *C_H_* are the myocyte and HEK cell capacitances, respectively (with units pF), and *G_gap_* is the intercell conductance (with units nS). *I_WT_* is the background HEK–WT current. *I_m_* is the total of the myocyte transmembrane currents, as defined by Paci et al. [24].

### 4.4. Immunocytochemistry

HEK cells were plated in 6-well glass-bottom plates (MatTek) and incubated for a minimum of 8 days at 37 °C, in 5% CO_2_. Existing medium was removed from the cells, and the culture was washed 3 times with PBS (without Ca^2+^ or Mg^2+^). The cells were fixed for 15 min at RT with 4% PFA and washed 3 times with 0.1% PBST (PBS + 0.1% Tween20). The cells were then permeabilized with 0.1% Triton X-100 diluted in PBS for 10 min at RT, and washed 3 times in PBST for 5 min each wash. The culture was then blocked for 60 min with 10% goat serum in PBS. The block solution was removed, and the cells were incubated overnight with 1:200 anti-connexin 43 antibody (rabbit) (Merck KGAA, Darmstadt, Germany) in PBST +10% goat serum. The cells were further washed 3 times in PBST at RT for 5 min for each wash and incubated for 1 h with anti-rabbit Ig-Alexa 647 at 1:200 dilution in PBST + 10% goat serum. The supernatant was removed and PBST with 1:1000 DAPI was added for 3 min. The cells were then washed 3 times with PBST before imaging. A control HEK culture was stained in the same manner, but without the anti-connexin 43 antibody. Stained cells were placed on an Olympus IX83 confocal microscope under 40× (N.A. 0.6, air objective) magnification. Illumination was switched to 637 nm laser light (Obis continuous wave laser, Coherent Inc, Glasgow, UK) with a 700/75 nm emission filter for visualization of Cx43. A 405 nm laser light (Obis continuous wave laser, Coherent Inc., Glasgow, UK) was used for visualization of the nucleus (DAPI). Minimal laser power was used in order to prevent significant photobleaching and a standard exposure time of 200 ms and an EM gain of 300 were used for the majority of images, but this was adjusted if required. Images were processed using ImageJ (NIH) to subtract the background and merge the channels.

### 4.5. Data Analysis and Statistics

All data were statistically analyzed on GraphPad Prism 6.0 (GraphPad Software, San Diego, CA, USA). Unpaired *t*-tests were performed for all data, except for when different experiments were compared (Figure 2 and Figure 4), in which case hierarchical statistics were performed: 2-way ANOVAs comparing the means of all experiments, and paired *t*-tests comparing each variable on one experiment. Data are shown as mean ± SEM, unless otherwise stated.

## Figures and Tables

**Figure 1 ijms-22-06621-f001:**
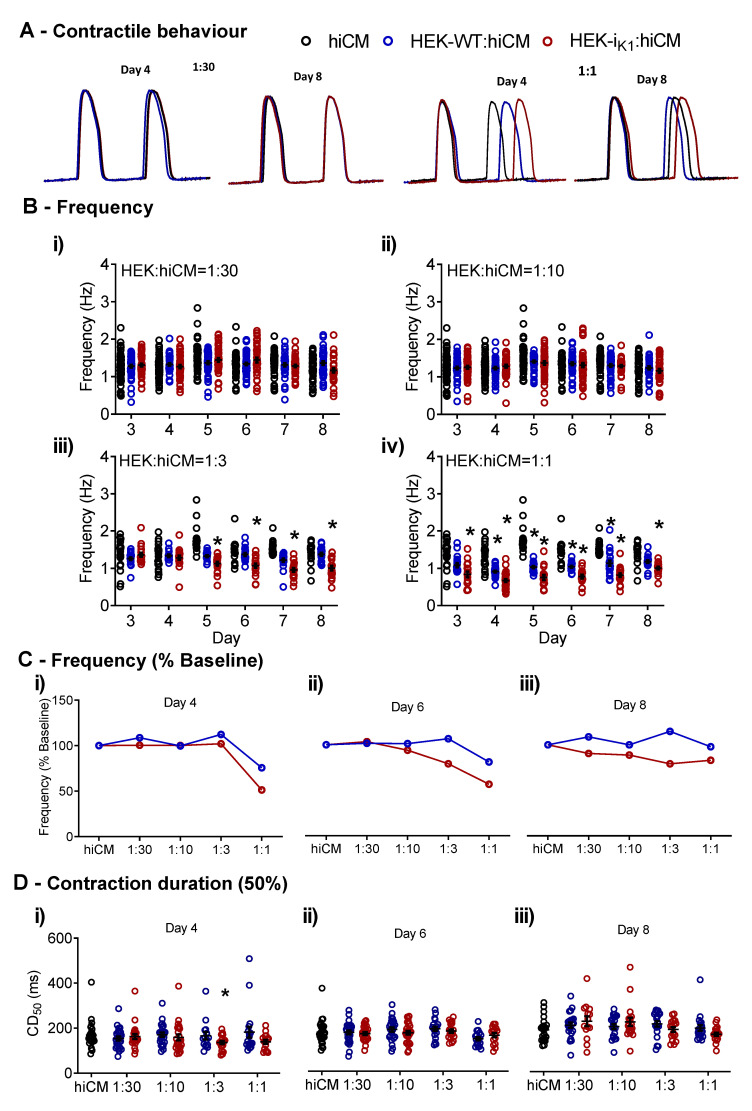
The effects of co-cultures of hiCMs with increasing densities of HEK, for both HEK-I_K1_:hiCM and HEK–WT:hiCM. (**A**) Example traces of the effects of 1:30 and 1:1 at days 4 and 8. (**Bi**) Frequency of spontaneous contraction at days 4 to 8 with the ratio 1:30; (**Bii**) frequency at the 1:10 ratio; (**Biii**) frequency at the 1:3 ratio; (**Biv**) frequency at the 1:1 ratio; (**C**) frequency as a percentage from baseline (standard hiCM culture) for the range of ratios. (**Ci**) Day 4; (**Cii**) day 6; (**Ciii**) day 8. (**D**) Contraction duration (CD50) on days 4, 6 and 8 comparing hiCM culture, HEK–WT:hiCM and HEK-I_K1_:hiCM. One-way ANOVA with Bartlett’s test, * *p* < 0.05, n = 51 wells.

**Figure 2 ijms-22-06621-f002:**
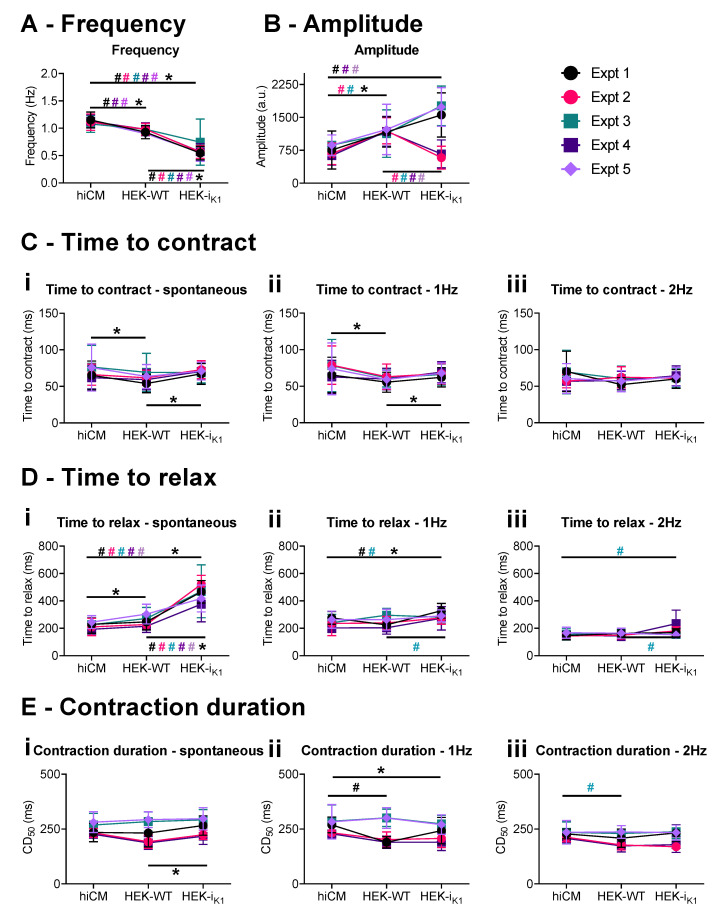
Cell movement effects of the 1:1 co-culture on day 4 in vitro in serum-free conditions, showing observations from individual experiments. Each plating is shown in a different color, and data are presented as mean+/−SD. (**A**) Frequency of spontaneous contraction. (**B**) Amplitude of the spontaneous contraction. (**C**) Time to contract at (**Ci**) spontaneous, (**Cii**) 1 Hz, and (**Ciii**) 2 Hz conditions. (**D**) Time to relax at (**Di**) spontaneous, (**Dii**) 1 Hz, and (**Diii**) 2 Hz. (**E**) This shows 50% of the contraction duration at (**Ei**) spontaneous, (**Eii**) 1 Hz, and (**Eiii**) 2 Hz. The comparison of each culture (hiCM, HEK–WT:hiCM and HEK-I_K1_:hiCM) within each experiment is shown as # in the respective experimental, and 2-way ANOVA was used for statistical analysis. The mean of all experiments was then compared using a paired *t*-test and this is shown as * (*p* < 0.05).

**Figure 3 ijms-22-06621-f003:**
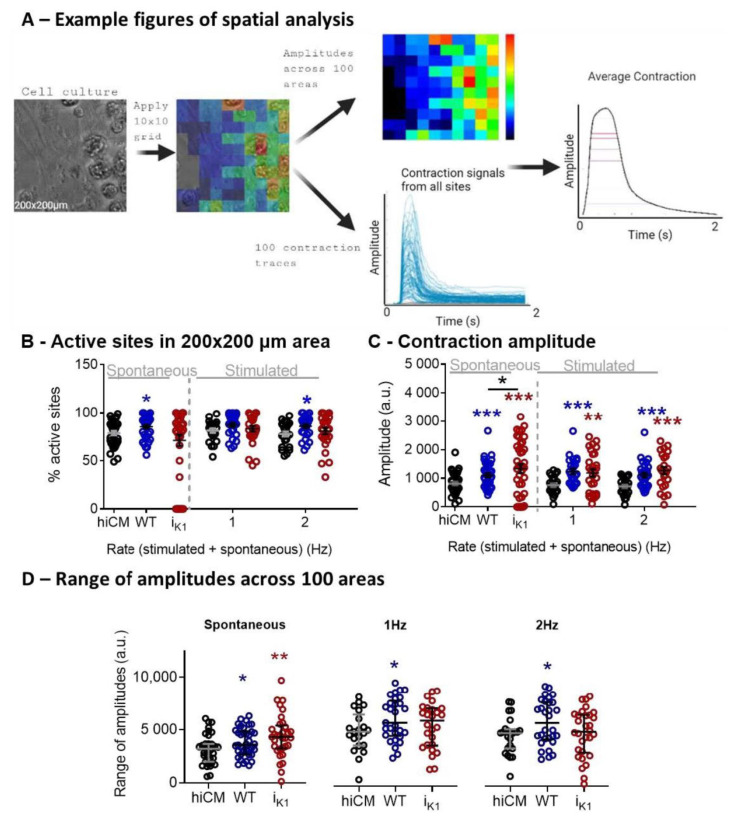
Spatial analysis of 1:1 HEK:CM on day 4 in culture in serum-free medium. (**A**) Example images obtained using spatial analysis of contractile motion. A heat map showing amplitudes recorded in a 200 × 200 µm area using a 10 × 10 grid were derived from the brightfield image (left). The colors represent the amplitude of the contraction in each square. Contraction signals from all sites and the average contraction are also shown. (**B**) Percentage of active sites representing contractile cells in a 200 × 200 µm area. (**C**) The amplitude of the contraction determined by pixel displacement. (**D**) Mean of range of amplitudes across a 10 × 10 grid. Unpaired *t*-test: HEK-I_K1_ vs. HEK–WT (black) and hiCM vs. HEK–WT (blue) or HEK-I_K1_ (red). *** *p* < 0.001, ** *p* < 0.01, * *p* < 0.05, n > 28 cells, 5 platings.

**Figure 4 ijms-22-06621-f004:**
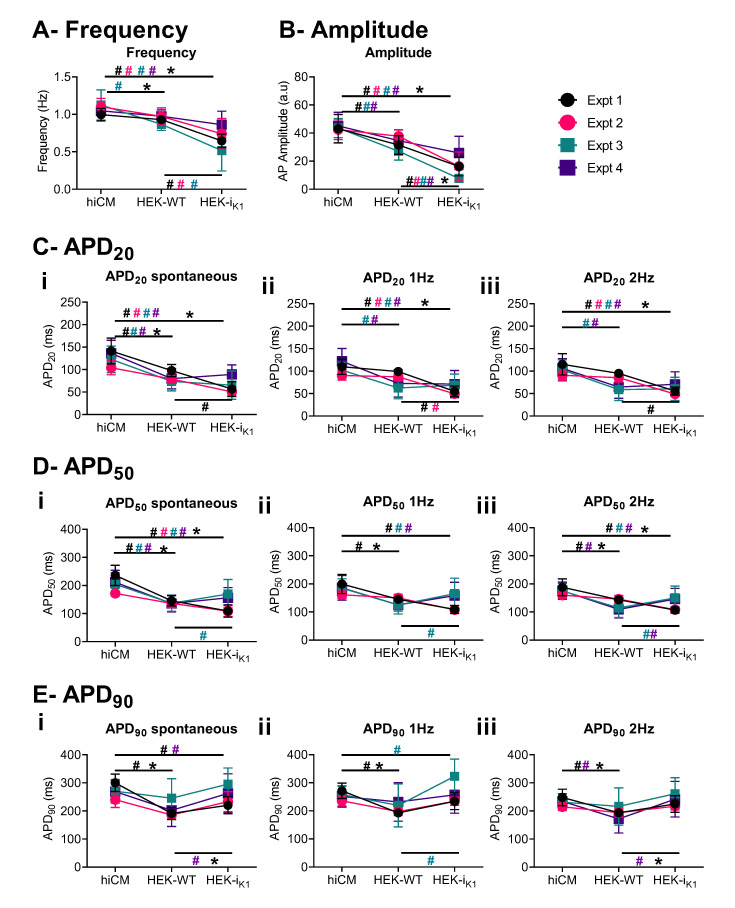
Electrophysiological effects of 1:1 co-culture on day 4 in vitro in serum-free conditions, showing the observations from individual experiments. Each plating is shown in a different color, and data are presented as mean+/−SD. (**A**) Frequency of spontaneous beating. (**B**) Amplitude of the AP. (**C**) APD20 at (**Ci**) spontaneous, (**Cii**) 1Hz, and (**Ciii**) 2Hz. (**D**) APD50 at (**Di**) spontaneous, (**Dii**) 1 Hz, and (**Diii**) 2 Hz. (**E**) APD90 at (**Ei**) spontaneous, (**Eii**) 1 Hz, and (**Eiii**) 2 Hz. Due to the small AP amplitude and low signal-to-noise ratio (SNR) (Appendix A) in HEK-I_K1_:hiCM, fewer data are available for APD90 at spontaneous rates. Comparison of each culture (hiCM, HEK–WT:hiCM and HEK-I_K1_:hiCM) within each experiment is shown as # in the respective experimental color, and 2-way ANOVA was used for statistical analysis. The mean of all experiments was then compared using a paired *t*-test and this is shown with * (*p* < 0.05).

**Figure 5 ijms-22-06621-f005:**
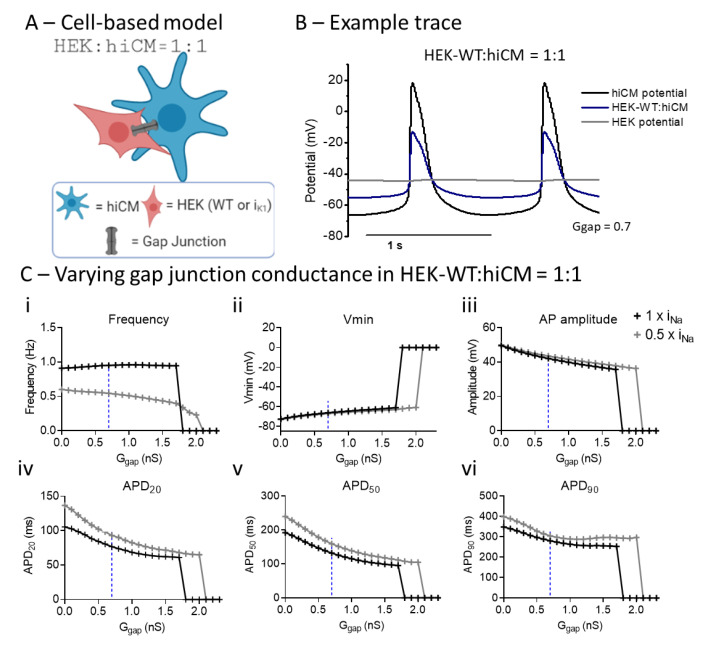
Computational modeling shows the effects of varying gap junction conductance (G_gap_) in a co-culture of hiCM with HEK–WT, and different Na^+^ channel expression. (**A**) Diagram of HEK:hiCM interaction used for computational model. (**B**) Example trace showing hiCM potential, HEK potential, and the average of the two. (**C**) Varying G_gap_ effects on frequency of spontaneous beating, hiCM’s Vmin, amplitude and APD. Simulations were done in regular I_Na_ (1 × I_Na_) and when only half (0.5 × I_Na_) was present.

**Figure 6 ijms-22-06621-f006:**
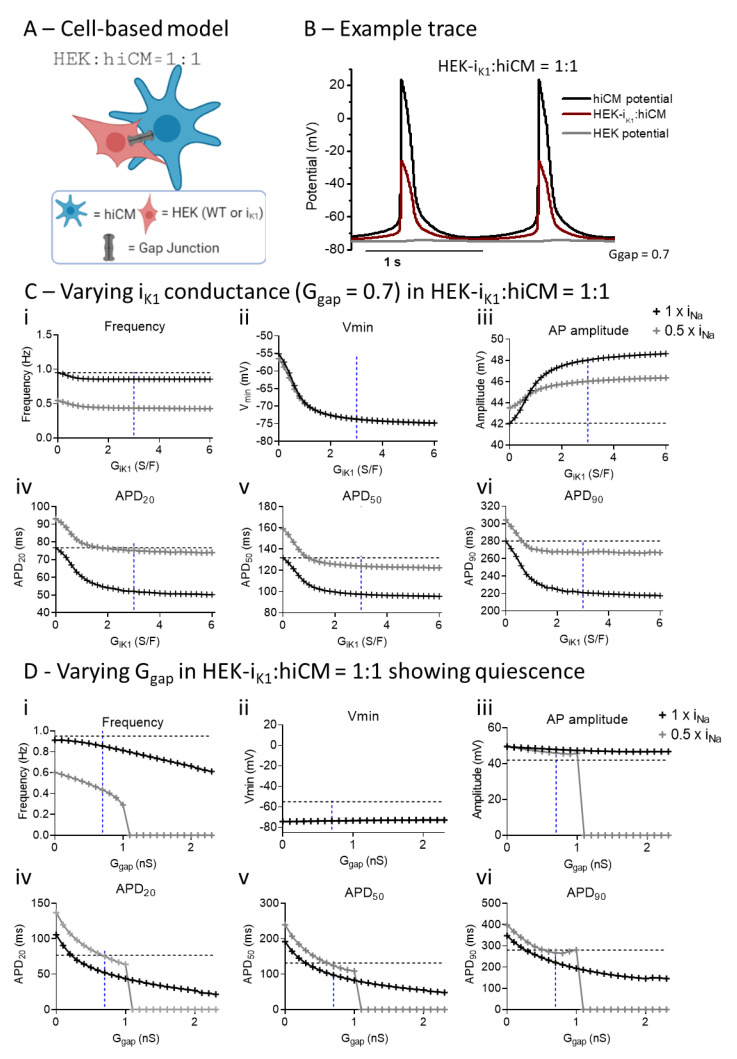
Computational modeling shows the effects of varying gap junction conductance (G_gap_) and I_K1_ conductance (G_IK1_) in co-cultures of hiCM with HEK-I_K1_ and different Na^+^ channel expressions. (**A**) Diagram of HEK:hiCM interaction used for computational model. (**B**) Example trace showing hiCM potential, HEK potential, and the average of the two. (**C**) Different parameters are affected by varying I_K1_ conductance in a full I_Na_ model, and a half INa. (**D**) Electrophysiology is affected by increasing gap junction conductance in a co-culture of hiCM with HEK-I_K1_.

## Data Availability

Not Applicable.

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
