# Peer review of "Electrophysiology of hiPSC-Cardiomyocytes Co-Cultured with HEK Cells Expressing the Inward Rectifier Channel"

_ijms, 2021, doi:10.3390/ijms22126621_

Round 1

Reviewer 1 Report

This manuscript reports on co-cultures of human iPSC-derived CMs and HEK cells overexpressing the Kir2.1 channel. The authors present a scientifically sound study with findings that are of importance to a general readership interested in the area of modeling contraction and electrophysiology with human iPSC-derived CMs.

Minor comments:

  • line 71: “Co-incubation with of hiCM with … ”
  • line 134: “Ik1:hiCM co-cultures (Supplementary Table 1)."
  • line 252: “APD20,_50 and 90 … “

Author Response

No response required.

Reviewer 2 Report

In this study, the authors investigate the impact of co-culturing hiPSC-derived cardiomyocytes (hiCMs) with HEK cells expressing IK1 current. Their results suggest that ratio of 1:1 is optimal to obtain viable co-cultures with electrical coupling shown by action potentials and cardiac sheet motions translated in contractile properties. Computational modelling explores the potential role of gap junctions in the co-cultured cardiac sheet. However, the data also indicate that such co-culture leads to partial loss of spontaneous contraction.

The topic is quite original and the engineered co-culture deserves interest. However, further key experiments are needed to reinforce the claims. The manuscript could be of much better quality as the way of writing displays clear lack of discipline (e.g. interpretations are often found in the result section and methods are enough described). The discussion is too light as well.

Major concerns:

Fig. 1: How did you measure these parameters? Was it using MuscleMotion and ImageJ? Please explain clearly how you proceeded as the method is light. Why the MM script is only mentioned for Fig. 3?

Please illustrate the results with some movies showing the difference in beat rate based on the time and cell ratio.

Fig. 1C and D: how the authors explain the change in frequency independently of the CD50? What about the contraction amplitude (i.e. indirect force)? Does decreased frequency correlates with increased contraction amplitude? Please show these data. What about the resting time between 2 cycles of contraction-relaxation? In fact, why the authors purposely selected the properties in figure 1? Please explain.

What are the evidences than IK1 is responsible for the frequency and CD50 change? Does IK1 causes structural/functional changes in hiCM? Please comment.

Fig. 2: please detail the pacing properties in the method. What was the duration and delay of stimulation? Was the 1:1 co-culture with HEK-IK1 able to respond to the pacing, in particular 2Hz, by increasing the beat frequency? I think this is a key question when assessing the impact of IK1 current.

What is the basal level of Kir2.1 in WT-HEK? Please comment.

To validate the role of IK1 current in the results of Fig. 2, I suggest to use a specific pharmacological inhibition of Kir2.1 ion channel such PA-6 at 1 and 2Hz to reverse the effects on contraction. Does IK1 inhibition reverse to WT-HEK properties?

Page 4, line 89: the interpretation of the results should be done in the result section. Please remove this paragraph. Same comment for page 6 (line 134) and 11 (line 260).

Fig. 5 and 6: Immunofluorescence and western blot experiments should be performed to reinforce the presence of gap junctions.

I suggest to evaluate the expression of HCN4 channel in the 1:1 co-culture. Can the funny current decrease explain the quiescent regions of the syncytium?

The discussion section should contain a paragraph on the spontaneous electrophysiological properties in hiCM.

Page 15, line 414: please explain how the mixture was performed before plating (time and experimental procedure).

Minor concerns:

Figure S3A : there is a mistake with “measurement”.

The supplementary material word file contains comments in tracking mode from the authors. Please correct.

Author Response

Reviewer 2

We appreciate the constructive point of view provided by the reviewer, we have tried to achieve all the suggestions in order to make a substantial improvement in the manuscript quality

The topic is quite original, and the engineered co-culture deserves interest. However, further key experiments are needed to reinforce the claims. The manuscript could be of much better quality as the way of writing displays clear lack of discipline (e.g. interpretations are often found in the result section and methods are enough described). The discussion is too light as well.

Thanks for the comments provided. We have revised the ms. according to these comments and hope the ms. is now satisfactory.

Major concerns:

Fig. 1: How did you measure these parameters? Was it using MuscleMotion and ImageJ? Please explain clearly how you proceeded as the method is light. Why the MM script is only mentioned for Fig. 3?

 Apologies, this should have been made clearer; yes, an implementation of the muscle motion algorithm was used. We have added text to the Methods section to clarify this.

Please illustrate the results with some movies showing the difference in beat rate based on the time and cell ratio.

We have added these movies to the supplementary data

Fig. 1C and D: how the authors explain the change in frequency independently of the CD50? What about the contraction amplitude (i.e. indirect force)? Does decreased frequency correlates with increased contraction amplitude? Please show these data. What about the resting time between 2 cycles of contraction-relaxation? In fact, why the authors purposely selected the properties in figure 1? Please explain.

The reviewer is correct, there are a series of contractile parameters that can be analysed and displayed, but we thought the CD50 was the most informative mechanistically. The lack of change in CD50 as frequency is reduced is exactly what we wanted to demonstrate, that frequency of beating rather than time-course of the contraction was the cause of the change in frequency. We have mde this clearer in the Discussion. The amplitude is an interesting parameter but in the context of this system/experiment it was not used for two reasons (i) rate-dependant changes in amplitude are minimal in hiCMs compared to adult tissue. (ii) the amplitude signal is the most variable parameter assessed with the MM algorithm because it depends on light level and detail in the image. For that reason, we do not report the amplitude routinely. We intended to simplify the graphical display by reporting only the most relevant parameter.  We hope this is a satisfactory explanation.

 What are the evidences than IK1 is responsible for the frequency and CD50 change? Does IK1 causes structural/functional changes in hiCM? Please comment.

The only difference between the two HEK cell lines is the presence of iK1. The slowing of spontaneous rate on when the HEK-Ik1 line was used is anticipated on theoretical grounds from the electrophysiological effects of increased Ik1 on spontaneous activity. Theory would indicate that AP duration may decrease slightly therefore we would anticipate no significant change in timecourse of the contraction if the AP dictated aspects of the Ca transient duration. We have clarified this line of thinking in the revised ms.

 Fig. 2: please detail the pacing properties in the method. What was the duration and delay of stimulation? Was the 1:1 co-culture with HEK-IK1 able to respond to the pacing, in particular 2Hz, by increasing the beat frequency? I think this is a key question when assessing the impact of IK1 current.

Details of the pacing/electrode system is now given in the Methods section of the ms. As described, the electrodes were 1mm diameter graphite electrodes ~ 4mm apart that were lowered into the media and a square-wave pulse of 6ms was used up to 40V was used. These details are given in the revised ms. The inactive areas (showing 0% active areas in Fig3B) could be paced by 1Hz and 2Hz stimulation as shown in Fig 3B. This is consistent with the reduced excitability observed only when co-culturing with HEK-IK1 cells. This is emphasised in the revised ms.

What is the basal level of Kir2.1 in WT-HEK? Please comment.

We have cited the electrophysiological study by Thomas and Smart (2005) showing the background electrophysiology of HEK293 cells. There is no evidence of inward rectification, in fact a weak outward rectification was observed. The lack of specific background conductances is one of the reasons this cell type is used to express ion channels (Thomas & Smart 2005) (Kirkton & Bursac 2011). This is emphasised in the revised form of the ms.

To validate the role of IK1 current in the results of Fig. 2, I suggest to use a specific pharmacological inhibition of Kir2.1 ion channel such PA-6 at 1 and 2Hz to reverse the effects on contraction. Does IK1 inhibition reverse to WT-HEK properties?

This is an excellent suggestion, and one we considered in these studies. We examined 100-300nM PA-6 on the electrophysiology of the iPSC-CM alone and failed to find effects consistent with iK1 inhibition (i.e. increased intrinsic frequency or shortening of APD), despite previous studies showing IK1 expression in this form of hiCM. Barium did affect spontaneous rate but also many other aspects of the electrophysiology, i.e. was clearly non-specific. Ivabradine reduced spontaneous rate but also prolonged the action potential indicating non-specific actions even as low as 1uM (Ana Costa, PhD thesis, University of Glasgow 2020). In summary, we regarded the pharmacological tools either insensitive or too non-specific to be useful in this study. Instead, our argument centres on the differential effects between HEK-WT and HEK-IK1 cells, i.e. due to the very specific difference between the two cell types. We hope this is a satisfactory response for the reviewer.

Page 4, line 89: the interpretation of the results should be done in the result section. Please remove this paragraph. Same comment for page 6 (line 134) and 11 (line 260).

We have moved this section as indicated by the reviewer.

 Fig. 5 and 6: Immunofluorescence and western blot experiments should be performed to reinforce the presence of gap junctions.

We have added examples of immunofluorescence images and details of the underlying methods to confirm the presence of connexin 43 in the HEK celllines used in this study. Others have shown the presence of Cx43 in HEK293 cells (Patel 2014) and in iPSC-CMs (Lu et al 2016). We have added clarifications to this effect throughout the revised ms. We hope these additions are sufficient to satisfy the reviewer’s comment.

I suggest to evaluate the expression of HCN4 channel in the 1:1 co-culture. Can the funny current decrease explain the quiescent regions of the syncytium?

We have studied ivabradine in iPSC-CM preparations and observed slowed spontaneous rate but not quiescent preparations with concentrations as high as 10uM. But as mentioned before, the drug lacks specificity at higher concentrations. It would also seem unlikely that by only changing the co-culture to IK1-expressing HEK cell that this would lead to a selective decrease in HCN4 in the adjacent iPSC-CM. We have added text to the discussion to discuss the option raised by the reviewer. We hope the reviewer agrees with the response.

The discussion section should contain a paragraph on the spontaneous electrophysiological properties in hiCM.

We have added a section on the spontaneous electrophysiology of iPSC-CMs.

Page 15, line 414: please explain how the mixture was performed before plating (time and experimental procedure).

Additional information has been provided in Methods as requested by the reviewer.

Minor concerns:

Figure S3A: there is a mistake with “measurement”.

Corrected

The supplementary material word file contains comments in tracking mode from the authors. Please correct.

Done

Thank you for these corrections.

Reviewer 3 Report

In this manuscript, Silva Costa and colleagues originally describe the possibility to introduce the stabilizing role of the IK1 in hiCMs thus improving therapeutic and pharmacological purposes. By co-culturing expressing IK1 HEK cells (HEK-IK1) with human-induced pluripotent stem cell-derived cardiomyocytes (hiCMs) in different ratios, they demonstrate that only the ratio of 1:3 HEK-IK1:hiCM (vs 1:30 or 1:10) significantly evidences a reduced spontaneous rate. hiPSC-derived CMs, indeed, are known to be lacking or poorly expressing IK1 component thus leading to a more depolarized membrane potential and spontaneous activity. This aspect in terms of “functional immaturity” is crucial to determine, together with the time of culture, an optimized physiological condition to investigate mechanisms of cardiac cellular disease and predict pharmacological approaches in such a human cellular model. Mechanical and electrophysiological parameters are evaluated in different co-culture conditions and computational modeling is proposed to support the hypothesis that co-culturing of hiCMs with a Human Embryonic Kidney (HEK) cell-line expressing the Kir2.1 channel (HEK-IK1) can generate an electrical syncytium with adult-like cardiac electrophysiology.

To this Reviewer, the manuscript is methodologically original but specific points should be clarified:

  • It is known that cardiac calcium handling and its modulation mechanism are crucial during EC-coupling in determining the cellular contraction properties. Since the cardiac calcium handling is finely tuned also at the sarcoplasmic level, I was wondering whether the authors considered the very different endoplasmatic calcium release in HEK cells and if this may interfere with normal cardiac EC coupling of hiPSC-CMs in their heterogeneous cocultures.

  • By using the changes in APD observed experimentally, the authors estimated the gap-junction conductance through the in silico model thus allowing the explanation of the APD changes themselves. From my point of view, it would be important to correlate this computational data with molecular expression level and cellular distribution of gap junctions proteins, in the different monolayers, to validate the Ggap estimation made by Paci’s model (e.g. PCR, immunostaining).

  • The physiological phenotype of hiPSC-CMs is heterogeneous both in terms of sub-populations of CMs (nodal, atrial, and ventricular cells) and in terms of maturation degree during differentiation protocol. IK1 electrical properties are different between atrial and ventricular CMs (Verkerk et al, 2017) and variation in AP waveform between hiPSC-CM is well known to be influenced also by the critical GK1 (Fabbri et al 2019), so I was wondering how the authors took this into account in their original work. Probably it would be better considered in the discussion section.

  • To this reviewer a little methodological curiosity relative to the aim of this work: what are the limitations or reasons not to consider a direct hiPSC-CMs transfection with Kir 2.1 gene (Vaidyanathan, R)? It would be appropriate to emphasize the concept in more depth in the discussion section.

Author Response

Reviewer 3

It is known that cardiac calcium handling and its modulation mechanism are crucial during EC-coupling in determining the cellular contraction properties. Since the cardiac calcium handling is finely tuned also at the sarcoplasmic level, I was wondering whether the authors considered the very different endoplasmatic calcium release in HEK cells and if this may interfere with normal cardiac EC coupling of hiPSC-CMs in their heterogeneous cocultures.

This is an interesting idea, our understanding was that the ER Ca release system in HEK cells was accessed via IP3 mediated release and in none of our studies were agonists of this system used so we did not anticipate this to be a confounding factor. If it were, we would expect it to be common between the two HEK cell lines. We hope that this is a sufficient explanation for the reviewer.

By using the changes in APD observed experimentally, the authors estimated the gap-junction conductance through the in-silico model thus allowing the explanation of the APD changes themselves. From my point of view, it would be important to correlate this computational data with molecular expression level and cellular distribution of gap junction proteins, in the different monolayers, to validate the Ggap estimation made by Paci’s model (e.g. PCR, immunostaining).

This is a good point, we have addressed this comment by including immunohiochemistry of the two HEK cell lines demonstrating Cx43 expression in both. We have added text in the discussion citing references that show the expression of Cx43. We cannot find a reference of Cx conductance measurements in HEK/iPSC-CM cell pairs, these are very specialised and difficult measurements. We hope by providing evidence of Cx43 expression and supporting literature this addresses this comment sufficiently.

The physiological phenotype of hiPSC-CMs is heterogeneous both in terms of sub-populations of CMs (nodal, atrial, and ventricular cells) and in terms of maturation degree during differentiation protocol. IK1 electrical properties are different between atrial and ventricular CMs (Verkerk et al, 2017) and variation in AP waveform between hiPSC-CM is well known to be influenced also by the critical GK1 (Fabbri et al 2019), so I was wondering how the authors took this into account in their original work. Probably it would be better considered in the discussion section.

This is a good point, and we have modified the discussion to reflect the reviewer’s point over variability of cell phenotype. When dealing with monolayers of 1000’s of electrically coupled cells we expect that cell-to-cell variation is averaged to an overall electrophysiological profile.

To this reviewer a little methodological curiosity relative to the aim of this work: what are the limitations or reasons not to consider a direct hiPSC-CMs transfection with Kir 2.1 gene (Vaidyanathan, R)? It would be appropriate to emphasize the concept in more depth in the discussion section.

We mention the shortcomings of alternative approaches in the Introduction, and in response to the reviewer’s comments we have developed these arguments in the revised Discussion. This included the methodological issues associated with the transfection of IK1 suggested by the reviewer. We hope this is sufficient to address this point.

Round 2

Reviewer 2 Report

The revised manuscript is now ready for acceptance. I have no further comment.